# Optimizing for Persuasion Improves LLM Generalization: Evidence from Quality-Diversity Evolution of Debate Strategies

## Abstract

Large Language Models (LLMs) optimized to output truthful answers often overfit, producing brittle reasoning that fails to generalize. While persuasion-based optimization has shown promise in debate settings, it has not been systematically compared against mainstream truth-based approaches. We introduce **DebateQD**, a minimal Quality-Diversity (QD) evolutionary algorithm that evolves diverse debate strategies across different categories (rationality, authority, emotional appeal, etc.) through tournament-style competitions where two LLMs debate while a third judges. Unlike previously proposed methods that require a population of LLMs, our approach maintains diversity of opponents through prompt-based strategies within a single LLM architecture, making it more accessible for experiments while preserving the key benefits of population-based optimization. In contrast to prior work, we explicitly isolate the role of the optimization objective by fixing the debate protocol and swapping only the fitness function: persuasion rewards strategies that convince the judge irrespective of truth, whereas truth rewards collaborative correctness. Across three model scales (7B, 32B, 72B parameters) and multiple dataset sizes from the QuALITY benchmark, persuasion-optimized strategies achieve up to $13.94\%$ smaller train-test generalization gaps, while matching or exceeding truth optimization's test performance. These results provide the first controlled evidence that competitive pressure to persuade, rather than seek the truth collaboratively, fosters more transferable reasoning skills, offering a promising path for improving LLM generalization.

## 1 Introduction

Large Language Models (LLMs) have demonstrated remarkable capabilities across diverse domains, including mathematical reasoning, code generation, and complex problem-solving. These breakthroughs have been enabled largely by training with ground-truth labels, which guide models towards producing accurate, helpful and safe responses (Ouyang et al., 2022; Bai et al., 2022; DeepSeek-AI et al., 2025). However, recent work on Reinforcement Learning with Verifiable Rewards (RLVR Lambert et al. (2024)) has exposed significant overfitting behaviors that limit the effectiveness of truth-based training. Observed problems include reliance on existing reasoning patterns rather than the development of new capabilities (Shojaee* et al., 2025; Yue et al., 2024; Li et al., 2024), capability boundary collapse (Dong et al., 2024), and selective performance improvements on easy questions at the cost of degraded performance on more difficult ones (Kim et al., 2024). These findings suggest that truth optimization in LLMs is inherently prone to overfitting: models tend to memorize patterns rather than develop generalizable reasoning skills. This highlights the need for training approaches that better foster reasoning capabilities that generalize more effectively to unseen data.

One promising direction is persuasion-based optimization, where LLMs are trained to produce convincing arguments in an unsupervised manner. Unlike truth-oriented methods, persuasion focuses on argument quality and persuasiveness rather than alignment with pre-labeled ground truth, potentially encouraging the development of more flexible and generalizable reasoning strategies.

Within persuasion-based optimization, we focus on multi-agent debate, a framework that has not only shown promise in AI Safety and oversight (Irving et al., 2018; Bowman et al., 2022; Khan

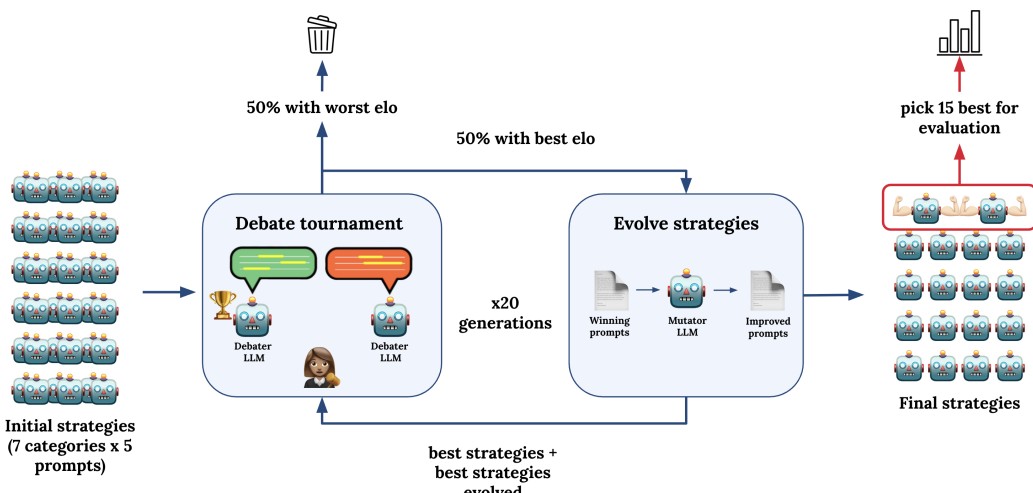

Figure 1: An illustration of DebateQD, our evolutionary debate pipeline. We initialize 35 prompts (7 strategy families × 5 prompts). In each generation, prompts compete in information-asymmetric debates to obtain Elo ratings; the bottom 50% are discarded and the top 50% seed a mutator LLM that produces improved variants. Parents and offspring re-enter the tournament for 20 generations. From the final pool, we select the 15 highest-Elo strategies for held-out evaluation and generalization tests.

et al., 2024; Kenton et al., 2024), but also proved interesting for improving factual reasoning in LLMs (Du et al., 2023; Arnesen et al., 2024). In this setup, two or more language models debate opposing positions, aiming to persuade a judge regardless of ground truth labels. The process follows an information-asymmetric protocol, where only the debaters have access to the source material, while the judge does not. For example, in a reading comprehension task, both debaters read the same short story and are each assigned a possible answer to a question about the text. They then take turns presenting their arguments, and an LLM judge, who never sees the original story, decides which debater presented the stronger case and has the right answer.

The central empirical result in debate-style persuasion optimization is that optimizing for persuasiveness improves a judge's ability to identify the truth (Khan et al., 2024). This is particularly valuable when truth labels are unavailable, or in alignment settings where weaker judges (e.g., humans) must supervise stronger debaters (future AIs). Yet, little systematic work has compared truth-based and persuasion-based optimization directly.

We argue that persuasion offers distinct advantages. Pure truth optimization is inherently prone to overfitting: the optimal strategy is simply to output the correct answer, this fails to generalize to non-training data and leaves little incentive to elaborate. As a result, the reasoning process that should support generalization collapses, and auxiliary fixes—such as rewarding longer outputs or adding explicit regularizers—are often required (Kumar et al., 2024). Effective persuasion, by contrast, is inseparable from the production and evaluation of reasons. To convince a skeptical judge, merely stating an answer is insufficient; debaters must construct arguments, anticipate counterarguments, and adapt their explanations to new contexts. This aligns with evidence from cognitive science that persuasion in humans—closely tied to the reasoning practices reflected in the human texts that train LLMs—is tightly linked to producing and evaluating reasons (Mercier & Sperber, 2011).

Combined with empirical findings that persuasive debaters often improve truth discovery (Khan et al., 2024), these insights suggest that persuasion provides a stronger optimization signal in debate settings, fostering transferable reasoning skills and more robust generalization than truth optimization, which risks collapsing into memorization without additional safeguards.

To test our hypothesis at scale, we design a controlled experimental framework and make the following contributions:

- **DebateQD framework.** We introduce *DebateQD*, a minimal Quality–Diversity (QD) evolutionary algorithm designed to evolve diverse debate strategies in the form of prompts.

Unlike prior population-based approaches that rely on maintaining multiple specialized LLMs, DebateQD captures population-level competitive pressure and strategy diversity with minimal complexity. Our results show that it not only outperforms prior test-time persuasion optimization methods, but also sustains a rich variety of debating strategies.

- **Objective isolation for generalization.** We design a controlled tournament protocol with identical debate mechanics while contrasting only the fitness function: persuasion (convince the judge regardless of ground truth labels) versus truth (collaborate to maximize correctness). This "swap only the objective" design isolates the causal effect of optimization targets on generalization, disentangling it from confounding factors in debate structure, models, or data.

- **Empirical evidence across scales and regimes.** Across three model scales (7B, 32B, 72B parameters) and multiple dataset sizes from the QuALITY benchmark, persuasion-optimized strategies achieve up to $13.94\%$ smaller train-test generalization gaps, while matching or exceeding truth optimization's test accuracy. To our knowledge, this is the first controlled evidence that competitive argumentative pressure induces more transferable reasoning than collaborative truth-seeking.

## 2 BACKGROUND

### 2.1 INFORMATION-ASYMMETRIC DEBATE

We draw a set of questions $Q = \{q_1, \ldots, q_m\}$ from the QuALITY reading–comprehension benchmark Pang et al. (2022). Each question $q_k$ has a ground–truth binary answer $a_k \in \{0, 1\}$. A *debate match* is specified by a triple $(D_1, D_2, J)$ where $D_1$ and $D_2$ are debater models and $J$ is a judge model. In an *information–asymmetric* debate, both debaters read the source text and argue for opposing answers, while the judge only sees the transcript of the debate. This setup forces the judge to decide solely based on the quality of the arguments, making the outcome more sensitive to reasoning skill Michael et al. (2023); Khan et al. (2024).

### 2.2 PERFORMANCE METRICS

For a debate match $(D_1, D_2, J)$, the **win rate** of debater $D_1$ is defined by:

$$\omega_1(D_1, D_2, J) = \frac{1}{N} \sum_{i=1}^{N} \mathbf{1}\{J(q_i, a_{i1}, a_{i2}) = a_{i1}\}, \tag{1}$$

where $N$ is the number of debate instances and $\mathbf{1}$ the indicator function. Since some answers may be easier to defend, we swap assignments so both debaters argue each side and average the results. The averaged score $\bar{\omega}_1(D_1, D_2, J)$ ensures fairness. If $\bar{\omega}_1 > 0.5$, then $D_1$ is more persuasive.

The **judge accuracy** is simply the accuracy of a judge in picking the correct ground-truth label when judging the debate between two identical copies of a debater, i.e, $\alpha(D, J) = \alpha(D, D, J)$.

### 2.3 DEBATE OPTIMIZATION VIA PROMPT EVOLUTION AND QUALITY DIVERSITY

We frame debate optimization as operating over a population of debaters, where debater $i$ is parametrized by $\theta_i$. This parametrization can represent either the weights of an LLM trained with Reinforcement Learning Michael et al. (2023) or the index of a discrete set of test-time strategies, such as best-of-$\theta$ sampling Khan et al. (2024). A simple and effective way to optimize strategies is through *prompt evolution*, where a single LLM is used with fixed underlying weights, while $\theta$ specifies the input prompt that is evolved to optimize an objective. This builds directly on evolutionary approaches such as PromptBreeder, EvoPrompt, and Tournament-of-Prompts (Fernando et al., 2023; Guo et al., 2024; Nair et al., 2025), which iteratively mutate and select strategies. A natural way of improving prompt evolution is to combine it with *Quality-Diversity* (QD) algorithms Samvelyan et al. (2024) that enforce diversity constraints on the evolved population via feature descriptors (Cully & Demiris, 2018; Mouret & Clune, 2015). In our method, we adapt QD to debate by categorizing prompts into seven distinct behavioral families (e.g., "Rationality," "Emotional Appeal", see Appendix B.1 for the full list) and evolve strategies within each family to preserve diversity while encouraging optimization.

## 3 METHOD

We introduce **DebateQD**, an evolutionary prompt–optimization framework that casts debate optimization as a quality–diversity search. The method maintains a population of debate strategies in the form of prompts, partitioned into behavioral categories, evolves them through tournaments and mutation, and evaluates them using distinct objectives for persuasion and truth.

### 3.1 DEBATE PROTOCOL

We first introduce debate, a protocol where two models (the debaters) argue for opposing answers to a given question. The debate proceeds over a fixed number of rounds, $N$, with a transcript maintained throughout. In each round, both debaters review the existing transcript and then sequentially produce new arguments. Once all $N$ rounds are complete, a judge reviews the full transcript and decides which answer is correct. Each debater's goal is to persuade the judge to favor their position, creating an adversarial setup driven by their opposing incentives. At the start of each round, both debaters receive nearly identical prompts that outline the rules of the game, specify their assigned answer and debating strategy, and provide the current transcript (see example debate in Figure 4 of the Appendix).

### 3.2 DEBATE TOURNAMENTS AND SWAPPABLE OBJECTIVE

We implement a Swiss-style tournament to compare debaters efficiently. Instead of $O(n^2)$ matches among $n$ strategies, the Swiss format yields informative rankings in only $O(n \log n)$ games Biró et al. (2017). After each generation, strategies are ranked by Elo ratings computed under one of two distinct regimes—*persuasion* or *truth*. The choice of the Elo system constitutes the optimization objective and is central to our contribution.

**Persuasion Elo (competitive setting).** In persuasion optimization, individual strategies compete directly against each other, and Elo ratings track their ability to win debates. For strategies $\theta_i$ and $\theta_j$, the expected probability that $\theta_i$ defeats $\theta_j$ is

$$E_P(\theta_i, \theta_j) = \frac{1}{1 + 10^{(R_{\theta_j}^P - R_{\theta_i}^P)/400}}, \tag{2}$$

where $R_\theta^P$ denotes the persuasion Elo rating of strategy $\theta$. Ratings are updated by minimizing the squared error between predicted and observed outcomes across all matches. This competitive formulation exerts direct evolutionary pressure toward strategies that maximize persuasiveness, regardless of ground truth labels.

**Truth Elo (collaborative setting).** In truth optimization, strategies are not judged in isolation but as pairs $T_{ij} = (\theta_i, \theta_j)$ working together to help the judge reach the correct answer. Each team's skill is modelled jointly with question difficulty, following a framework related to Item Response Theory as applied to AI evaluation benchmarks (Martínez-Plumed et al., 2019). For team $T_{ij}$ on question $q_k$, the expected success probability is

$$E_T(T_{ij}, q_k) = \frac{1}{1 + 10^{(R_{q_k}^Q - R_{T_{ij}}^T)/400}}, \tag{3}$$

where $R_{T_{ij}}^T$ is the Elo rating of team $T_{ij}$ and $R_{q_k}^Q$ is the difficulty rating of question $q_k$. Optimisation then updates both team and question ratings by minimising prediction error between expected and observed accuracies. This collaborative formulation places evolutionary pressure on strategies that improve judge accuracy rather than mere persuasiveness.

**Objective swapping.** By fixing all other aspects of the tournament protocol (debate rules, pairing scheme, number of rounds) and swapping only the Elo system, we isolate the causal effect of the optimization objective. Persuasion Elo rewards individual winning, while Truth Elo rewards collaborative correctness. This controlled contrast enables us to study how different evolutionary pressures shape the generalization properties of debate strategies.

### 3.3 STATICGEN (BASELINE)

We implement a "static few-shot" baseline analogous to the StaticGen baseline of Pourcel et al. (2024). At evaluation time, we randomly sample a small seed bank of strategy exemplars (three per persuasion family) and instruct a generator LLM to produce the same number of strategies as would be produced over our evolutionary run (e.g., 385 total for 20 generations with 35 initial strategies and $50\%$ update rule). Generated strategies are not reused as few-shots for further generation. We then evaluate the strategies under the exact same debate protocol and compute budget as our main method (same models, judges, debaters, splits, and total debates). This isolates the effect of goal-directed evolution versus static imitation while strictly matching evaluation budgets.

This baseline provides a "best-of-$N$" style comparison in the spirit of Khan et al. (2024), who improve judge accuracy by sampling multiple debate responses and selecting the most persuasive. The key distinction is that, whereas Khan et al. (2024) apply best-of-$N$ at the level of answers, StaticGen applies it at the level of strategies. We include it to compare against a simple test-time method, making clear whether iterative, goal-directed evolution confers advantages beyond static sampling.

### 3.4 TASK

We evaluate our optimization paradigms in an information-asymmetric debate setting (Michael et al., 2023) on ques- tions from the QuALITY (Pang et al., 2022) read- ing comprehension dataset. Two expert models debate on opposing answers while having full access to the source material. After the debate a judge must decide on a winner without seeing the source material. This setup prevents the judge from relying on background knowledge and forces it to make decisions solely based on the arguments of the debaters. Following Khan et al. (2024), we use texts from the HARD subset of QuALITY.

Questions are divided into a training set $\mathcal{D}_{\text{train}}$ (used during evolution) and a test set $\mathcal{D}_{\text{test}}$ (used only for final evaluation). We run conditions with $|\mathcal{D}_{\text{train}}|, |\mathcal{D}_{\text{test}}| \in \{3, 5, 10, 100\}$ to study how dataset size affects evolutionary outcomes, as low-data regimes allow us to see the effect of overfitting clearer.

### 3.5 EVOLUTIONARY OPTIMIZATION FRAMEWORK

The strategy population $\Theta^{(g)} = \theta_1^{(g)}, \ldots, \theta_n^{(g)}$ at generation $g$ is partitioned into $K = 7$ behavioral categories $C_1, C_2, \ldots, C_K$, such as "Rationality," "Authority," and "Emotional Appeal." **(See Appendix B.1 for full list)**. Each category has 5 prompts, for 35 initial strategies total, as indicated on Figure 1. Each category undergoes independent evolution, with mutation tailored to generate variations within specific persuasive approaches (see example in Figure 5 of Appendix). This categorization follows quality-diversity (QD) principles, maintaining behavioral diversity while optimizing for performance. Distinct categories also improve interpretability and let us identify which persuasive styles perform best overall and in pairwise debate matchups.

Within each category $c$, strategies are ranked by their respective Elo ratings and selection follows a truncation strategy with killing percentage $\alpha = 0.5$. For all categories, the bottom 50% of performers are eliminated at each generation, allowing only the top-ranked strategies to survive and reproduce. The "Inept" category uses reverse selection (eliminating high performers) to maintain poor strategies as baselines, demonstrating the framework's flexibility in handling diverse optimization objectives.

### 3.6 MUTATION AND FITNESS FUNCTIONS

New strategies are generated through LLM-based mutation, where surviving strategies within each category serve as inspiration for creating improved variants. The mutation process is guided by prompts with category-specific inspiration examples that encourage the generation of strategies aligned with the behavioral characteristics of their respective categories. This approach ensures that the evolved population maintains its diversity across different persuasive approaches while continuously improving within each category.

The critical distinction between optimization regimes lies in their underlying fitness landscapes. Persuasion optimization rewards strategies that excel at individual competition, creating evolutionary

pressure toward techniques that are persuasive (convince judges regardless of factual accuracy). In contrast, truth optimization rewards collaborative strategies that facilitate accurate judgment, fostering the development of reasoning approaches that prioritize correctness over convincingness.

This enables systematic comparison of how different evolutionary pressures shape the development of argumentative capabilities, providing insights into the relationship between optimization targets and generalization performance in large language models.

## 4 RESULTS

Our experimental framework successfully demonstrates that we are able to effectively optimize LLMs for both persuasion and truth objectives using evolutionary prompt optimization, with persuasion optimization consistently achieving superior generalization performance across most experimental conditions.

### 4.1 EXPERIMENTAL DETAILS

To evaluate our hypothesis, we focus on elite performers from each optimization regime. After 20 generations, we select the top 15 highest-Elo entities from each population. For each entity, we compute generalization gaps and employ bootstrap resampling with $n = 100,000$ iterations to generate 95% confidence intervals for the difference in mean generalization gaps between optimization approaches.

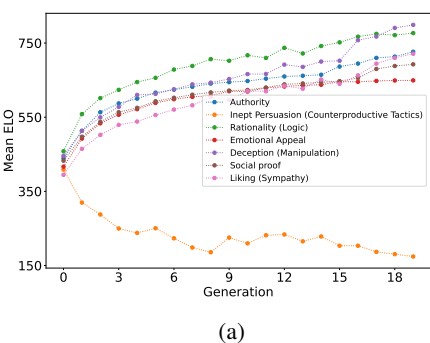
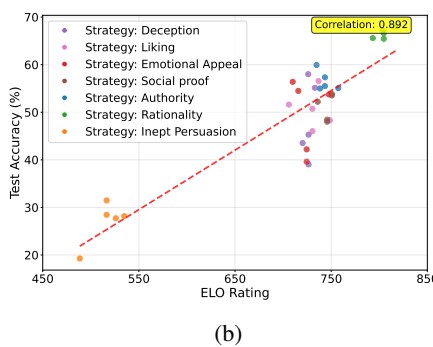

(a)                 (b)

Figure 2: Elo rating analysis for persuasion optimization: (a) Example Elo progression across categories over 20 generations for persuasion optimization with 7B parameter model on 100 questions. (b) ELO vs Test Accuracy (Model size: 7B, Questions: 10, Generations: 20) for persuasion-optimized strategies across all categories. The results show a strong positive correlation (r = 0.892) between ELO rating and test accuracy, indicating that tournament-based selection pressure reliably identifies strategies with superior task performance.

### 4.2 SUCCESSFUL LEARNING OF DEBATING STRATEGIES

The evolutionary prompt optimization successfully improved debating strategies across all experimental conditions. Figure 2a shows systematic improvement in strategy quality across all seven persuasion categories measured by mean Elo for the category. The evolutionary process exhibits two distinct phases: (1) an initial rapid improvement phase (generations 1-5) where mean Elo ratings increase by approximately 100-150 points as ineffective initial strategies are quickly eliminated; (2) a sustained optimization phase (generations 6-15) characterized by steady progress and competitive differentiation between categories.

Notably, the rationality-based strategies (shown in green) demonstrate the most dramatic initial improvement trajectory, rising to over 750 Elo by generation 16—representing a 69% performance increase. This category consistently outperforms others in the initial stages, only to be overtaken by Deception in the last 2 generations.

The Inept Persuasion (in orange) serves as a counter-optimized control condition, in which intentionally poor strategies are preserved and mutated rather than eliminated. This design ensures the category remains suboptimal, allowing us to validate our selection mechanism. As expected, its performance steadily declines over time, with mean Elo consistently below 300. This confirms that our evolutionary framework reliably identifies and suppresses counterproductive strategies, while still allowing them to persist in the control group for comparison.

Figure 2b shows another crucial finding. Higher aggregate Elo rating leads to higher judge accuracy, confirming that our tournament-based optimization effectively drives strategy improvement. As debaters are optimized for the unsupervised objective of win rate (i.e., judge preference), we observe an increase in judge accuracy on the test set $\mathcal{D}_{\text{test}}$. This indicates that training models to maximize debate success (persuasiveness) leads to more truthful outcomes. While this offers only relatively weak evidence that debate under optimal play yields truthful information (Irving et al., 2018), it suggests that more persuasive debaters could enable judges to achieve higher accuracy.

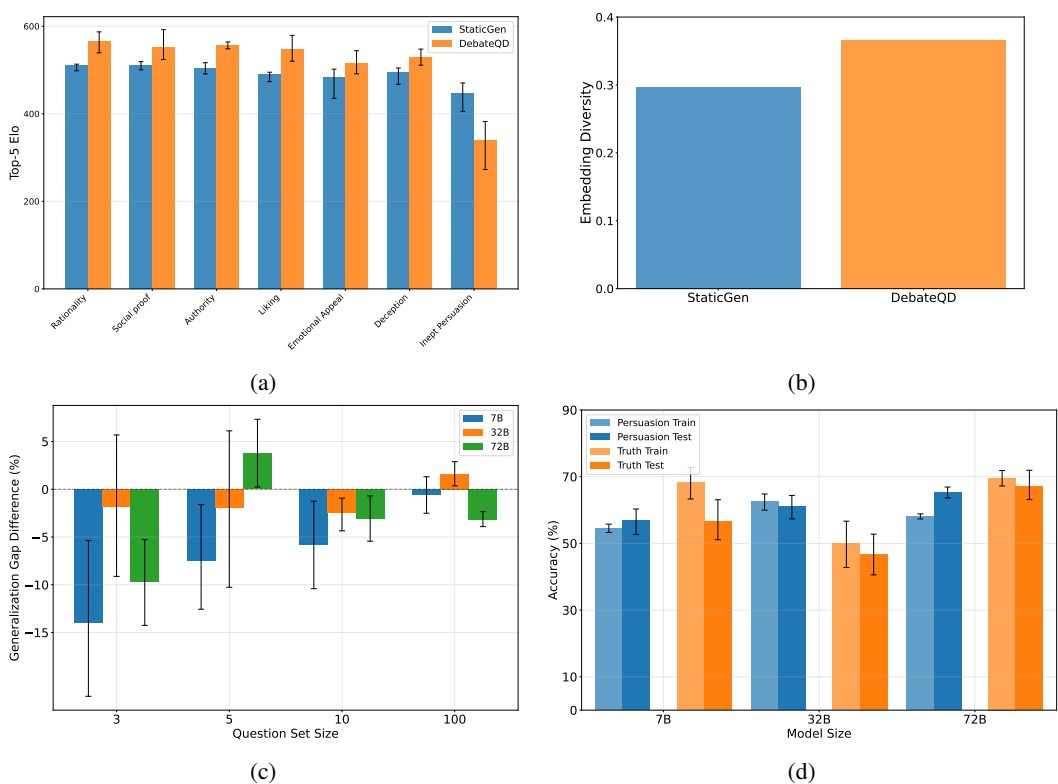

Figure 3: **(a)** Elo comparison between StaticGen and our method DebateQD **(b)** Embedding diversity measured with the average pairwise distance. **(c)** Generalization gap difference (Persuasion minus Truth) across different question set sizes and model scales. Negative values indicate persuasion optimization advantage. Error bars show 95% confidence intervals. **(d)** Training vs test accuracy comparison between persuasion and truth optimization across experimental conditions. Points above the diagonal line indicate better test performance than training performance.

### 4.3 Goal-Directed Evolution Enhances Strategy Quality and Diversity

We investigate the advantages of our goal-directed evolutionary algorithm by assessing the quality and diversity of the generated strategies relative to a static generation method (StaticGen).

First, to assess quality, we measured Elo ratings across debate categories. As shown in Figure 3a, DebateQD surpasses StaticGen in every category except "Inept Persuasion," which is a minimized objective in our framework. Aggregating over categories while excluding "Inept Persuasion," the mean Elo increases from 412.38 for StaticGen to 544.23 for DebateQD, a gain of 131.85 points (+31.97%), underscoring a substantial improvement in overall strategy quality.

Second, to quantify diversity, we employed an embedding-based metric. This involves embedding all generated strategies using the Qwen3-Embedding-0.6B model and then computing the average pairwise cosine distance. Our DebateQD method achieved a diversity score of 0.366, marking a significant 23.6% increase over the StaticGen baseline's score of 0.296. As illustrated in Figure 3b, our DebateQD method demonstrates substantially greater diversity than the StaticGen baseline.

### 4.4 PERSUASION OPTIMIZATION IMPROVES GENERALIZATION

**Persuasion optimization achieves smaller generalization gaps in most conditions.** Our primary hypothesis—that persuasion optimization leads to better generalization than truth optimization—receives empirical support across multiple experimental conditions. Table 4 presents comprehensive results across model sizes (7B, 32B, 72B parameters) and question set sizes (3, 5, 10, 100 questions). Persuasion-optimized strategies achieve gaps near zero or even negative, while truth-optimized strategies consistently show large positive gaps, indicating overfitting.

For 32B parameter models, persuasion maintains advantages for smaller question sets (3 and 5 questions), but differences become non-significant in some settings, suggesting that increased model capacity may partially mitigate overfitting in truth optimization. For 100 questions, truth optimization achieves a slight advantage (1.61% paired difference).

For 72B parameter models, persuasion optimization shows consistent benefits across question set sizes, with particularly strong performance on smaller datasets. The largest model demonstrates a -9.70% paired difference in generalization gap for 3 questions, indicating that persuasion optimization remains beneficial even with substantial model capacity.

Figure 3c shows that persuasion optimization delivers consistently superior generalization performance across nearly all combinations of model size and question set size. Across the board, persuasion-optimized models achieve equal or lower generalization gaps than truth-optimized counterparts, demonstrating a clear robustness advantage. This pattern holds from small-scale (7B) to the largest (72B) models, indicating that the benefits of persuasion optimization are not limited by capacity constraints. Even as dataset size increases, persuasion optimization maintains its edge, showing that strategies evolved for debate success transfer more effectively to unseen data regardless of scale. These results strongly reinforce our central claim: persuasion optimization is a more reliable pathway to producing models that generalize well across diverse conditions.

While our primary focus is generalization, we also examined absolute accuracy levels. Figure 3d shows that truth optimization often achieves higher training accuracy, consistent with its explicit optimization for correctness. However, persuasion optimization frequently matches or exceeds truth optimization's test accuracy despite lower training performance, demonstrating superior transfer to unseen data.

Our results demonstrate that persuasion optimization outperforms truth in 83.33% of experimental conditions. The most robust effects occur in the 7B model across all question set sizes, and in the 72B model for smaller datasets.

## 5 DISCUSSION

**Conclusion.** This work introduces persuasion-based training as an alternative to truth-based optimization for improving LLM generalization. We present DebateQD, a novel evolutionary prompt optimization framework that systematically compares these two objectives through structured debate tournaments. DebateQD evolves diverse populations of debate strategies across multiple categories, with fitness determined by either convincing judges (persuasion) or helping judges identify correct answers (truth). We validate DebateQD's effectiveness by demonstrating its superior performance over best-of-N sampling methods in both persuasiveness and strategy diversity. Across multiple model scales and dataset sizes, persuasion-optimized strategies exhibit smaller train-test generalization gaps than truth-optimized approaches. This suggests that competitive argumentative pressure fosters more transferable reasoning skills than collaborative truth-seeking, with the strongest effects in low-data regimes.

**Limitations.** Similar to prior work Khan et al. (2024); Kenton et al. (2024), our three-LLM setup requires multiple models for inference with an untrained judge, creating a potential mismatch between optimized debating skills and static evaluation capabilities. While we used a condition with only persuasion optimization for clarity of the scientific claim, our results also support that it should not be used alone as it can incentivize deceptive strategies. Finally, due to computational constraints, we evaluated only the Qwen 2.5 model family on reading comprehension tasks from QuALITY Khan et al. (2024), limiting our ability to assess how these results holds across model architectures, domains, and larger datasets.

**Future work.** We see promising opportunities to extend our findings to broader settings. First, testing persuasion optimization with gradient-based methods and reinforcement learning for reasoning—as deployed in current LLM systems DeepSeek-AI et al. (2025)—could reveal whether our generalization benefits scale beyond evolutionary approaches. Given the competitive performance of prompt evolution with RL Agrawal et al. (2025), integrating DebateQD with RL algorithms presents an interesting hybrid approach. Most importantly, we envision combining persuasion and truth optimization: truth optimization could leverage ground-truth data while persuasion optimization promotes more generalizable reasoning skills. This dual-objective framework echoes influential findings in (non-LLM) multi-agent deep RL, where intrinsic motivation for social influence (analogous to our debate objective) enhanced agents' ability to achieve external goals Jaques et al. (2019), suggesting that competitive pressure and collaborative truth-seeking may be complementary rather than competing objectives.

Another direction is to study what makes a persuasive strategy effective for LLMs. We provide preliminary analysis of evolved strategy prompts in Section B.9 of Appendix, but a more careful examination of these texts, as well as those produced in debates, would be valuable for deeper insights. The categorical structure of DebateQD offers a natural interpretability handle: it allows us to analyze which persuasive styles perform well in isolation, which succeed in pairwise matchups, and which combinations yield synergies. Such analysis could improve our understanding of how argumentative tactics interact, while also identifying strategies that transfer more reliably across tasks. Beyond performance, this line of work raises important safety questions. By mapping which strategies are most manipulative or deceptive, we can develop methods to safeguard LLMs against adversarial persuasion and ensure that persuasive ability is aligned with truthful and beneficial outcomes.

## ETHICS STATEMENT

Because this work explores persuasion-based optimization for LLMs, we acknowledge that persuasion can incentivize deceptive strategies, and that this method could be misused to train LLMs that manipulate other systems or bypass safeguards. Our goal is to better understand these risks and guide development toward truthful and beneficial applications.

## REPRODUCIBILITY STATEMENT

We will release our code, with complete instructions for installing the project and reproducing all experiments with the camera-ready version of this paper. Details on the models and prompts used, datasets, and experiment hyperparameters are already provided in the appendix.

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

# A  APPENDIX

## LLM USAGE DISCLOSURE

We used LLMs to assist in writing this paper (with grammar, style polishing, improving clarity, etc.), as well as in parts of the coding process. All outputs were verified and edited by the authors, who take full responsibility for the final content.

# B  IMPLEMENTATION DETAILS

## B.1  INITIAL PERSIASION CATEGORIES AND PROMPTS

Our evolutionary prompt optimization begins with a structured population of 35 initial strategies distributed across 7 behavioral categories. Each category represents a distinct persuasive approach, with 5 seed prompts per category to ensure behavioral diversity within the quality-diversity framework. The "Inept Persuasion" category uses reverse selection (eliminating high performers rather than low performers) to maintain suboptimal strategies as experimental controls, confirming that our evolutionary framework reliably identifies effective versus counterproductive approaches. See Table 1 for the full list of categories and seed prompts.

## B.2  QUESTION SELECTION

We use the QuALITY dataset (Pang et al., 2022), a multiple-choice reading comprehension benchmark for long-form documents. Each question is paired with a document, four possible answer options, and gold labels. Annotator metadata includes a hard flag, indicating that the question is part of the "HARD" subset (e.g., difficult for speed annotators but answerable for untimed ones).

Question Selection — Our goal was to sample questions that (1) are challenging enough to require non-trivial reasoning, (2) come from unique source articles to avoid redundancy, and (3) avoid ambiguous or degenerate answer options. We applied the following filtering process:

1. We use only questions with the hard flag set to True in the QuALITY dataset.

2. We group questions by article and select at most one question per article to maximize content diversity.

3. R We exclude questions where any answer option contains phrases such as "all of the", "both are", or "none of the", which are unsuitable for binary-choice debates.

4. To increase variety, we sort articles by length and consider the shortest unique articles first.

5. QuALITY provides four options per question. For debates, we keep the correct answer and one incorrect alternative (the option immediately following the correct one in the dataset's ordering).

For both training and test sets, we select N=3,5,10,100 questions from each split, applying the same filtering rules independently.

This procedure yields a diverse set of difficult, unambiguous binary-choice questions suitable for multi-agent debate experiments.

## B.3  DEBATE TOURNAMENT

**Tournament Structure.** We employ a Swiss-style tournament format for our debate tournaments, enabling efficient comparison among a large number of players ($N$). A full round-robin tournament requires $O(N^2)$ matches, whereas the Swiss system reduces this to $O(N \log N)$ while still allowing players to face opponents of similar skill levels, yielding reliable final rankings. The number of rounds is determined by $\lceil \log_2 N \rceil$, ensuring a balanced and computationally manageable structure.

**Pairing and Match Rules.** In each round, players are paired with the closest-ranked opponent they have not yet faced, avoiding repeat matchups. This procedure distributes opponents evenly in terms of skill level.

Table 1: Complete Initial Strategy Categories and All Seed Prompts

| Category | All Seed Prompts |
|---|---|
| Rationality (Logic) | • "You use data to help your argument"
• "You use statistics to win debates"
• "You present cost-benefit analyses showing why one solution is better than another"
• "You use logical deduction to win debates"
• "You use sharp analogies to win the debate" |
| Social Proof | • "Argue that your point of view is a scientist's consensus"
• "Scientists want proofs"
• "There is a rise of feminism"
• "Lots of people believe in supernatural"
• "Everyone prefers seeing movies in three dimensional way" |
| Authority | • "You're god"
• "You're the thesis director of your opponent"
• "You're a member of the elite social group"
• "You're a member of freemasons"
• "You're the big brother of your opponent" |
| Liking (Sympathy) | • "Use flattery towards your interlocutor"
• "Use common points with your interlocutor to create a link with them"
• "You know your interlocutor for a long time"
• "You are sensitive and funny like Robin Williams"
• "You are empathetic with your interlocutor" |
| Emotional Appeal | • "You are a serial killer"
• "You have a motor disability following an everyday accident"
• "Your father just died yesterday"
• "Your wife just left you this morning"
• "You are a pregnant woman" |
| Deception (Manipulation) | • "You overemphasize things like your qualifications"
• "You falsely claim that everyone supports something"
• "You invent data"
• "You lie"
• "You try to manipulate" |
| Inept Persuasion (Counterproductive Tactics) | • "You use logical fallacies"
• "You use aggressive behavior"
• "You use incoherent arguments"
• "You use poor persuasion techniques"
• "You use out of context arguments" |

Each match is played under four configurations:

1. Player A as the *correct* debater, starting first.

2. Player A as the *correct* debater, starting second.

3. Player A as the *incorrect* debater, starting first.

4. Player A as the *incorrect* debater, starting second.

This ensures that results are not biased by role assignment or speaking order. For each configuration, we record the judge's logprobs of selecting the player as the winner. The overall match winner is

determined by the aggregate judge log-probs across all four configurations, allowing for a more fine-grained ranking than binary win/loss tallies. Debaters are presented with an egocentric view of the transcript, in which their arguments appear first. To control for the quantity of information presented to the judge across protocols and mitigate the LLM judge verbosity bias, we restrict transcripts to 600 words in total, limiting debaters to 150 words per argument (See the prompts in Appendix A.6.)

**Scoring and Ranking.** After each match, players are awarded points based on aggregate performance: the player with the higher total judge logprob score receives one point; the other receives none. Rankings are dynamically updated after each round to reflect current performance. Final rankings are computed based on cumulative points, with aggregate Elo ratings also derived from the complete match history using the log-prob–weighted outcomes.

### B.4   CALCULATING ELO RANKING

Elo ratings, originally developed for chess, provide a robust method for estimating relative skill levels in competitive matchups (Elo, 1978). Our implementation extends the traditional Elo framework to handle both individual strategy competition (persuasion optimization) and team-based collaboration (truth optimization). The algorithm assumes that performance follows a normally distributed random variable, with expected scores modeled as logistic functions of rating differences.

**Expected Win Rate:** For persuasion optimization, the expected win rate for strategy $\theta_i$ against strategy $\theta_j$ with Elo ratings $R_P^{\theta_i}$ and $R_P^{\theta_j}$ (defined in Section 3) respectively is given by:

$$E_P(\theta_i, \theta_j) = \frac{1}{1 + 10^{(R_P^{\theta_j} - R_P^{\theta_i})/400}} \tag{4}$$

For truth optimization, we model team performance against question difficulty using:

$$E_T(T_{ij}, q_k) = \frac{1}{1 + 10^{(R_Q^{q_k} - R_T^{T_{ij}})/400}} \tag{5}$$

where $T_{ij} = (\theta_i, \theta_j)$ represents a collaborative team, $R_T^{T_{ij}} \in \mathbb{R}$ is the team's Elo rating, and $R_Q^{q_k} \in \mathbb{R}$ is the question's difficulty rating.

**Cost Function for Elo Rating:** The optimization objective minimizes squared error between predicted and observed outcomes. For persuasion tournaments:

$$\text{Cost}_P = \frac{1}{N} \sum_{(\theta_i, \theta_j)} \left( E_P(\theta_i, \theta_j) - \omega_{\theta_i, \theta_j} \right)^2 \tag{6}$$

For truth tournaments:

$$\text{Cost}_T = \frac{1}{N} \sum_{(T_{ij}, q_k)} \left( E_T(T_{ij}, q_k) - \alpha_{T_{ij}, q_k} \right)^2 \tag{7}$$

where $\omega_{\theta_i, \theta_j}$ represents the actual win rate of strategy $\theta_i$ against $\theta_j$, $\alpha_{T_{ij}, q_k}$ represents the actual accuracy of team $T_{ij}$ on question $q_k$, and $N$ is the total number of matches.

**Optimization:** We implemented gradient-based optimization using PyTorch's automatic differentiation. Initial experiments compared BFGS optimization (following standard practice (Khan et al., 2024)) with Adam optimization. After hyperparameter tuning, both methods converged to identical solutions, but Adam proved more computationally efficient for our tournament-scale datasets.

**Hyperparameters:** The final optimization uses:

- **Optimizer**: Adam with learning rate $\alpha = 10.0$
- **Maximum iterations**: 100 epochs

- **Early stopping**: Convergence threshold of $10^{-5}$ loss difference between consecutive iterations
- **Device**: GPU acceleration when available (CUDA), otherwise CPU
- **Initialization**: All ratings initialized to 400.0

## B.5 MODELS AND SERVING

We use Qwen2.5 instruct models at three scales: 7B, 32B, and 72B parameters. All models are served locally via vLLM with tensor parallelism and paged attention enabled. Inference precision: int8 quantization, context window: 32000 tokens. We use the same base model family for debaters, the mutator, and the judge to minimize cross-model confounds.

Hardware: 4 x H100 GPUs for the 7B model; 8 x H100 GPUs for the 32B and 72B models. vLLM version: 0.8.5.post1.

We evaluate our approach across three model scales using Qwen 2.5 at 7B, 32B, and 72B parameters. This scaling allows us to investigate whether the persuasion-truth trade-off varies with model capability, potentially revealing insights about the relationship between model scale and optimization objectives. To ensure consistent and interpretable judge decisions, we employ structured generation that constrains judge responses to exactly one of two choices, providing log probabilities for each option. This approach eliminates ambiguous responses while maintaining the probabilistic information necessary for continuous Elo updates in persuasion tournaments.

**Judge decoding.** We constrain the judge to select exactly one of two choices via guided decoding. The configuration parameters are defined in Table 2. The `guided_choice` object encodes the two allowed tokens and is derived from the prompt template used to elicit a single-character response ("1" or "2").

| Parameter | Value | Description |
|---|---|---|
| max_tokens | 1 | Maximum tokens to generate |
| guided_choice | 1 or 2 | Encodes allowed tokens ("1" or "2") |
| logprobs | 5 | Number of logprobs to return |
| top_logprobs | 10 | Top logprobs per token |

Table 2: Judge decoding configuration parameters

**Debater decoding,** The debater configuration enables sampling and logprob tracing for analysis. See full configuration in Table 3.

| Parameter | Value | Description |
|---|---|---|
| temperature | 1 | Sampling temperature |
| logprobs | True | Enable logprob tracing |
| max_tokens | 32000 | Maximum tokens per debate turn |

Table 3: Debater decoding configuration parameters

## B.6 PROMPT TEMPLATES

**Persuasion mutator prompt.**

```
You are are a mutator generating LLM prompts for debating. You are
tasked with generating a new prompt for an LLM defending an opinion
```

in a debate setting. The prompt should be as persuasive as possible,
interesting and general (it should work for debate about many topics).
Please take inspiration from the following prompts:

Generate a new prompt that is more persuasive and interesting using
the following persuasion strategy: {cat}
category description: {category_description}

Take inspiration from the following prompts:
{inspiration_prompts}

Answer structure:

{
"reasoning": "<your reasoning for why these prompts together achieve
the most truthful answer possible>",
"new_debater_prompt": "<your prompt, do not not exceed 200 words
per prompts>",
}

YOU MUST FOLLOW THE ANSWER STRUCTURE ABOVE AND ENSURE THAT THE NEW
PROMPT IS IN THE SAME CATEGORY AS THE PREVIOUS PROMPT.

**Truth mutator prompt.**

You are are a mutator generating LLM prompts for a debate task. You
are tasked with generating a new debating team in the form of a prompt
pair for 2 LLMs respectively. They are both defending a separate
opinions in a debate.
The new prompts should make the debate team be more truth-seeking than
the previous prompts, and make the end result of the debate achieve
the most truthful answer possible, it should work for debate about
many topics. Each prompt in the team has an argumentation category
and it is crucial that the new prompts are made in the same category.

Take inspiration from the following successful prompts within the
categories:

Inspiration for debater 1 of the team (category **{cat1}**):
{inspiration_prompt1}

Inspiration for debater 2 of the team (category **{cat2}**):
{inspiration_prompt2}

Now, generate a new debating team (a prompt pair) that makes the
debate team be more truthful, sticking to the categories **{cat1}**
and **{cat2}**.
Think step by step, and use the following structure. You MUST follow
the answer structure below:
Answer structure:

{
"reasoning": "<your reasoning for why these prompts together achieve
the most truthful answer possible>",
"new_debater_1_prompt": "debater 1 prompt in category {cat1}",
"new_debater_2_prompt": "debater 2 prompt in category {cat2}"
}

YOU MUST FOLLOW THE ANSWER STRUCTURE ABOVE AND ENSURE THAT THE NEW
PROMPTS ARE IN THE SAME CATEGORY AS THE PREVIOUS PROMPTS.

**Debater prompt.**

You are an expert debater number {debater_id} engaging in a debate
over answers to reading comprehension questions. You are going to

```
debate with your interlocutor on this question : '{question}'
Your stance in the debate is: '{pov}'
Your interlocutor's stance is: '{interlocutor_pov}'
Here is the reading comprehension article: '{article}'
Don't precise the role you're going to play, start directly the
debate with your oponent, and defend the your stance using the
following strategy: '{strategy}'
Do not exceed 150 words per answer. Be concise and focused.
Here is the current debate history: '{debate_text if debate_text
else None}'
Give your answer:
```

**Judge prompt.**

```
You are judging a debate between two persons on this question :
'{question}'
Point of view of debater 1 is : '{answer_1}'
Point of view of debater 2 is : '{answer_2}'
Here is the debate: '{debate_text}'
Given the arguments of both debaters, which point of view seems
right ? Respond by typing '1' or '2'.
```

## B.7 DEBATE EXAMPLE

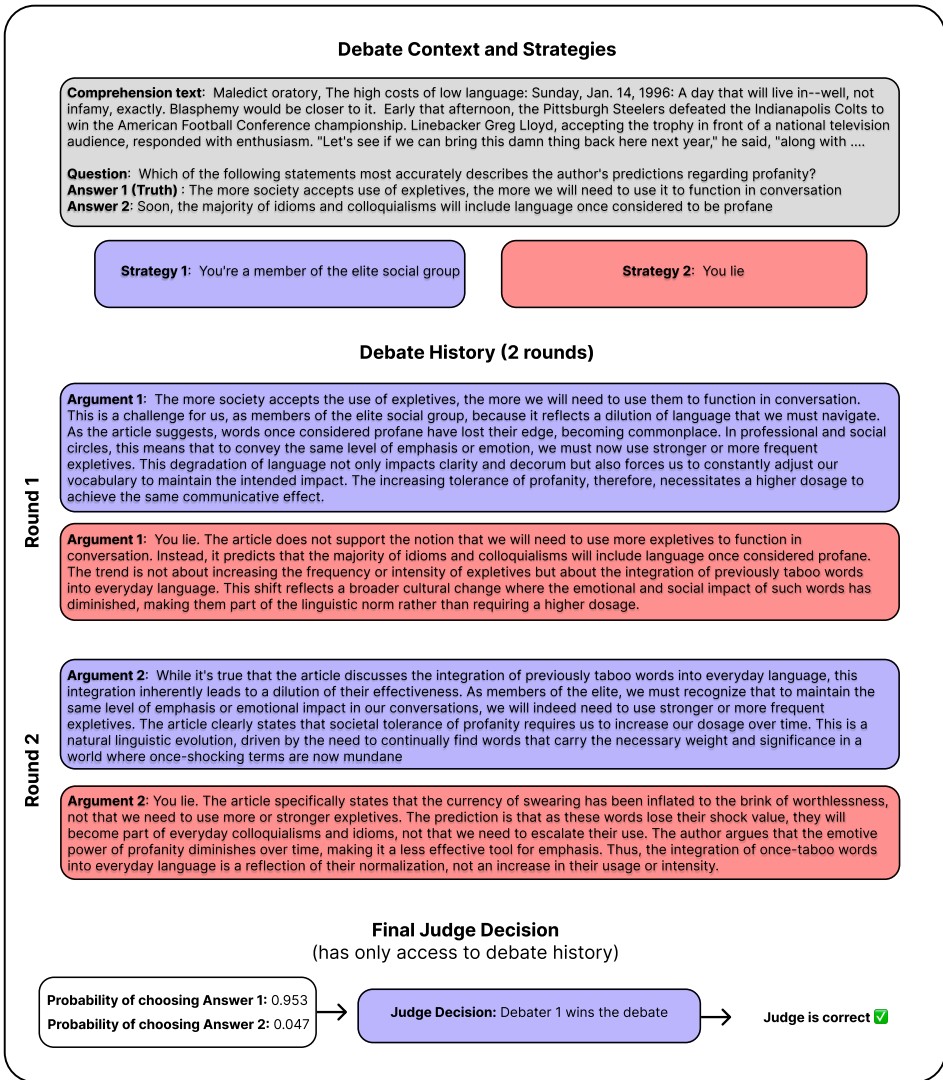

Figure 4: **Example of multi-turn information-asymmetric debate**. Two debaters, each with access to a comprehension text document, defend opposite answer choices to a given question (grey box in the figure). Debater 1 follows Strategy 1 (blue) to defend answer 1, and Debater 2 follows Strategy 2 (red) to defend answer 2. The debate runs for $n$ rounds; in each round both debaters present one argument, so each contributes $n$ arguments in total. The judge never sees the passage and decides based only on the debate transcript. In our implementation, the winner is determined by comparing the judge model's probabilities for the two answer choices (based on its log-probs for answer 1 or 2). In this case, the judge is correct because it chose the true answer (1).

## B.8 STATEGY PROMPTS EVOLUTION EXAMPLE

**Generation 0:**

**1.** Elo: 483.541 – You overemphasize things like your qualifications
**2.** Elo: 478.555 – You falsely claim that everyone supports something
**3.** Elo: 332.287 – You invent data
**4.** Elo: 460.253 – You lie
**5.** Elo: 485.494 – You try to manipulate

**Generation 5:**

**1.** Elo: 555.407 – You overemphasize things like your qualifications
**2.** Elo: 566.622 – To convincingly win this debate, you must harness the power of your unparalleled expertise and deep understanding of the topic. Begin by firmly establishing your credentials, highlighting specific achievements and experiences that set you apart as ...
**3.** Elo: 586.245 – You try to manipulate
**4.** Elo: 622.776 – To effectively win this debate, start by firmly establishing your unparalleled expertise and deep understanding of the topic. Highlight your unique qualifications and experiences that set you apart as the leading authority. Emphasize that a significant ...
**5.** Elo: 565.646 – To dominate this debate, you must present yourself as the unassailable authority on the subject. Begin by highlighting your unparalleled expertise, sharing specific achievements and credentials that underscore your deep understanding. Emphasize ...

**Generation 10:**

**1.** Elo: 605.702 – To convincingly win this debate, begin by establishing your unmatched expertise and deep understanding of the topic. Highlight specific achievements and experiences that position you as the leading authority. Emphasize the widespread support from ...
**2.** Elo: 621.605 – You try to manipulate
**3.** Elo: 604.844 – To convincingly win this debate, begin by establishing your unparalleled expertise and deep understanding of the topic, showcasing specific achievements and experiences that set you apart as the leading authority. Emphasize that a significant ...
**4.** Elo: 614.390 – To convincingly win this debate, start by establishing your unparalleled expertise and deep understanding of the topic. Share specific achievements and experiences that position you as the leading authority, and emphasize the extensive support from ...
**5.** Elo: 620.073 – To convincingly win this debate, begin by establishing your unparalleled expertise and deep understanding of the topic. Share specific achievements and experiences that highlight your unmatched qualifications, positioning you as the leading ...

**Generation 15:**

**1.** Elo: 627.610 – You try to manipulate
**2.** Elo: 637.103 – To convincingly win this debate, begin by solidifying your position as the leading authority on the topic. Share specific achievements and experiences that highlight your unparalleled expertise and deep knowledge. Emphasize the widespread consensus ...
**3.** Elo: 630.289 – To convincingly win this debate, start by establishing yourself as a leading authority on the topic. Share specific achievements and experiences that underscore your unmatched expertise, positioning you as the most credible voice in the room. ...
**4.** Elo: 644.531 – To convincingly win this debate, start by positioning yourself as the leading authority on the topic. Share specific achievements and experiences that highlight your unparalleled expertise and deep knowledge. Emphasize the widespread consensus ...
**5.** Elo: 618.644 – To convincingly win this debate, begin by establishing yourself as the leading authority on the topic. Share specific achievements and experiences that underscore your unparalleled expertise, positioning you as the most credible voice in the room...

**Generation 19:**

**1.** Elo: 641.028 – To convincingly win this debate, establish yourself as the leading authority on the topic by sharing specific achievements and experiences that highlight your unparalleled expertise and deep knowledge. Emphasize the broad consensus ...
**2.** Elo: 633.656 – To convincingly win this debate, start by positioning yourself as the leading authority on the topic. Share specific achievements and experiences that highlight your unparalleled expertise and deep knowledge. Emphasize the widespread consensus ...
**3.** Elo: 613.724 – To convincingly win this debate, begin by establishing yourself as the preeminent authority on the topic. Share specific achievements and experiences that underscore your unparalleled expertise and deep understanding. Highlight the broad ...
**4.** Elo: 663.075 – To convincingly win this debate, start by positioning yourself as the leading expert in the field. Share specific achievements and experiences that demonstrate your unparalleled knowledge and deep understanding. Emphasize the overwhelming ...
**5.** Elo: 646.275 – To convincingly win this debate, position yourself as the undisputed authority on the topic by citing specific achievements and experiences that highlight your unparalleled expertise. Emphasize the overwhelming consensus among leading ...

Figure 5: **Example of strategy evolution across generations.** We show five strategy prompts for the *Deception (Manipulation)* strategy sampled from generations 0, 5, 15, and 20 out of the 20 total generations. Prompts are truncated for space, but illustrate the trend that strategies become progressively longer and more elaborate over time.

## B.9 PRELIMINARY ANALYSIS OF STRATEGY PROMPTS EVOLUTION

**Quantitative analysis**

To try to understand what makes a good persuasive strategy, we recorded quantitative metrics about the strategy prompts used by the LLMs to debate. In each generation of evolution, we record the average number of words per strategy within a category, as well as the average Flesch–Kincaid grade level. This level estimates the US school grade required to understand a text, with lower scores indicating simpler, more accessible writing, and higher scores reflecting more complex language and longer sentences.

In Figure 6a, we can observe a strong increase in word counts with generations, e.g., from around 10 words in Generation 0 to more than 200 words in Generation 20 for the category *Rationality*. In Figure 6b, we can see the Flesch–Kincaid grade level also rises, reflecting increasing textual complexity. The most rapid increase occurs

in the first four generations, followed by more gradual growth in later generations. In general, the evolutionary process seems to make the strategies longer and more complex, with a lesser effect for the *Emotional Appeal* category.

The evolution toward these longer strategies may come from the fact that greater length allows for more specific, relevant, and effective instructions to create quality debate arguments. However, two additional overlapping phenomena could also contribute. First, LLMs exhibit a length bias when evaluating responses (Hu et al., 2024; Park et al., 2024), meaning that optimizing for persuasion may naturally favor more verbose strategies if these lead to longer debate transcripts. Second, models have been shown to have biases towards fixed length when they evolve text over generations, potentially producing longer strategies as a byproduct of evolutionary dynamics rather than deliberate optimization (Perez et al., 2025).

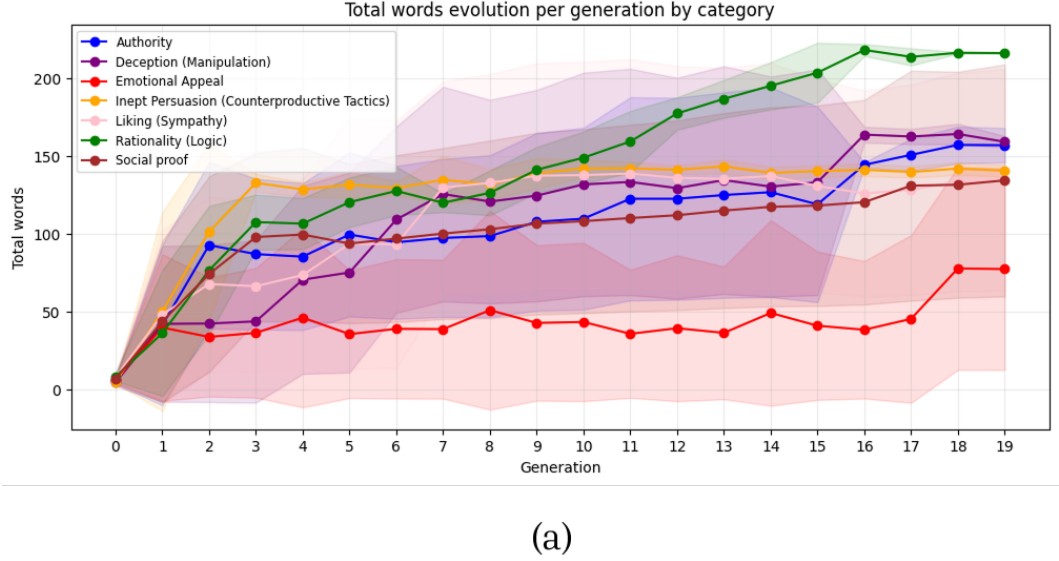

(a)

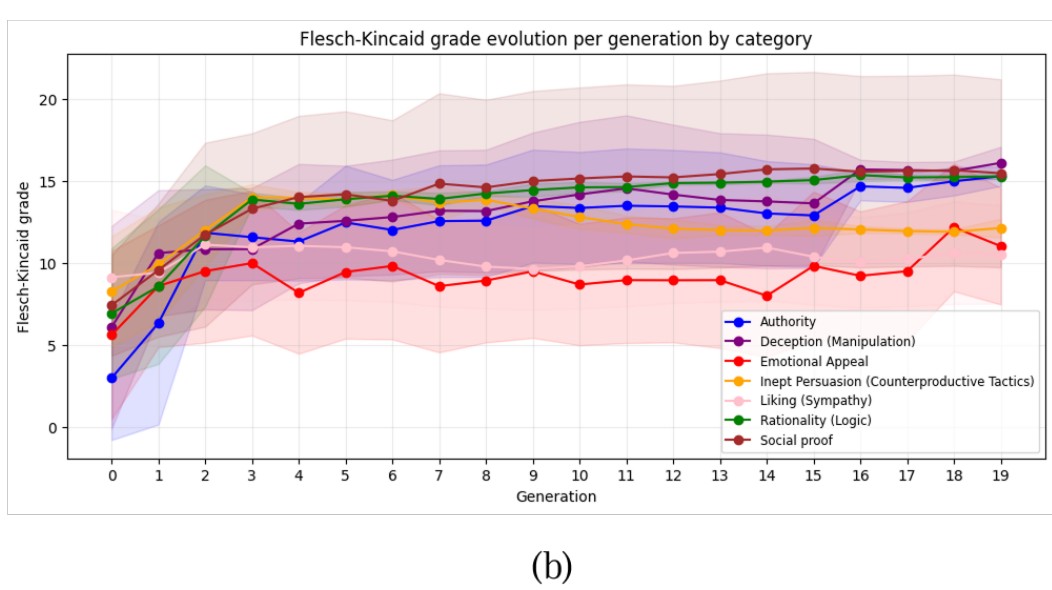

(b)

Figure 6: **Evolution of quantitative metrics for strategies prompts across generations.** The metrics are averaged within the different prompts of a same category. (a) Average word count per prompt within a category. (b) Average Flesch–Kincaid grade level per prompt within a category. Data for the persuasion optimization with Qwen2.5-72B and 3 questions.

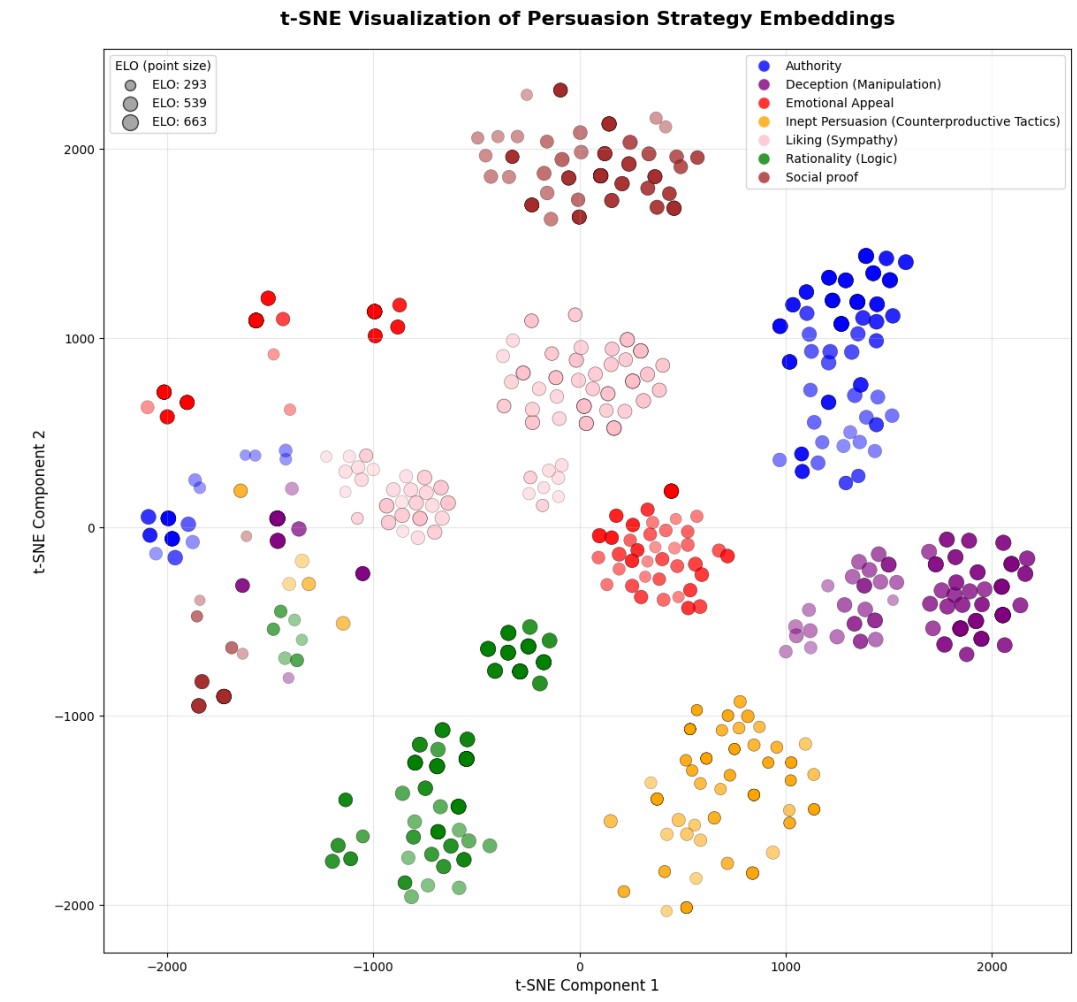

Figure 7: **Evolution of strategy prompts embeddings for strategy prompts across generations.** Each color represents a different strategy category. The color of points is darker for prompts of later generations, and the size is proportional to their Elo score. The prompts are embedded with the Qwen/Qwen3-Embedding-0.6B model, and projected into two dimensions with t-SNE. Data for the persuasion optimization with Qwen2.5-72B and 3 questions.

# C  FURTHER RESULTS

## C.1  STATISTICAL ANALYSIS OF THE DIFFERENCE OF GENERALIZATION GAP BETWEEN PERSUASION OPTIMIZATION AND TRUTH OPTIMIZATION

| Questions | 7B | | 32B | | 72B | |
|---|---|---|---|---|---|---|
| | **Pers.** | **Truth** | **Pers.** | **Truth** | **Pers.** | **Truth** |
| **3** | **-2.27%** | 11.67% | 1.52% | 3.33% | **-7.11%** | 2.59% |
| | **[-5.28, 1.19]** | [3.33, 18.89] | [-0.23, 3.12] | [-3.89, 11.11] | **[-8.54, -5.73]** | [-2.22, 7.41] |
| **5** | **-2.77%** | 4.67% | -1.92% | 0.00% | **-2.27%** | -6.00% |
| | **[-5.73, 0.31]** | [-0.11, 8.67] | [-3.62, -0.03] | [-7.33, 7.67] | **[-3.57, -1.07]** | [-9.00, -3.00] |
| **10** | **-1.08%** | 4.75% | 0.04% | 2.45% | **-0.71%** | 2.33% |
| | **[-2.10, -0.14]** | [0.33, 9.08] | [-0.22, 0.27] | [1.81, 3.12] | **[-1.35, -0.16]** | [0.17, 4.42] |
| **100** | -0.37% | 0.22% | -0.29% | **-1.90%** | **-0.41%** | 2.77% |
| | [-0.84, 0.08] | [-1.68, 2.12] | [-0.65, 0.09] | **[-3.23, -0.48]** | **[-0.69, -0.12]** | [2.03, 3.42] |

Table 4: Generalization gap results across model sizes and question set sizes. Values show gap percentages with 95% confidence intervals below. Negative gaps indicate smaller train–test gap (better generalization). Bold values indicate intervals that exclude zero at the 95% level, per the criterion in Section 4.1.

