# OpenReview forum: "Optimizing for Persuasion Improves LLM Generalization: Evidence from Quality-Diversity Evolution of Debate Strategies"
_ICLR.cc/2026/Conference — Submitted to ICLR 2026_

### Official Review · Reviewer_rS4P · 2025-11-01

**Soundness:** 3
**Presentation:** 3
**Contribution:** 2
**Rating:** 4
**Confidence:** 4

**Summary:**

The paper explores how different objectives (such as persuasion vs. truth-seeking) affect the generalizability of LLM debates. Using a multi-round debate tournament setup and an evolutionary Quality–Diversity algorithm, the authors evolve debate strategies and evaluate them on reasoning benchmarks. They find that persuasion-oriented strategies, while less accurate on training data, generalize better to unseen questions.

**Strengths:**

The paper is methodologically creative. The experimental design provides a scientific way to quantify argument quality. The authors’ analysis of behaviors and the link between persuasion and generalization offers novel insights for the readers in this domain.

**Weaknesses:**

Comment order: descending importance

Comment 1. Persuasion evolution appears to select longer, more elaborate prompts. This can raise two risks: (i) verbosity bias in LLM judging, and (ii) optimization pressure toward manipulative tactics that could mislead weaker judges. Restricting the length limit (as the authors stated in page 15) is not enough.
The authors can check the following:
(1) Normalize or penalize per-turn and total transcript length and re-estimate the persuasion–truth generalization gap. Doing so, the authors can quantify how much does “verbosity” contribute to the generalization gap reduction.
(2) Conceive a small sample where “intuitive but false” option is deliberately tempting. The authors can check whether deceptive strategies can hurt truth discovery when the debate is made intentionally confusing.

Comment 2. How did the authors construct seven behavioral families used in the QD framework? If they are manually chosen, it needs more justification. Further, several include ethically problematic personas (“you’re god,” “serial killer”). This mixes stylistic and moral dimensions, making it unclear whether the observed improvement comes from diverse reasoning styles or simply from judge bias toward certain tones (coming from specific personas). It also risks unsafe debate styles.
The authors can try the following:
(1) Replace these fixed categories with continuous style embeddings (e.g., discourse-level feature vectors). If persuasion still outperforms truth strategies, that would show robustness beyond the handcrafted taxonomy.
(2) Redefine behavioral families using neutral rhetorical language (e.g., evidence-first, balanced-counterargument, uncertainty-aware) and check whether the generalization gap persists.
(3) I would also like to see a win-rate matrix between all family pairs.

Comment 3. Without further retraining, the authors can evaluate evolved persuasion strategies directly on unrelated long-context reading datasets. I would expect some degradation in performance, but it would serve as a good benchmark whether the strategies possess cross-domain generalizability.

Comment 4. Using the same LLM model as a judge when evaluating texts generated by debate agents can be problematic as the judge may prefer patterns that match its own text generation style. Again, instead of re-training everything, the authors can perform a quick check by replacing the judge with a different LLM. Alternatively, the authors can perform independent human evaluations on a selected sample to ensure that the same generalization gain persists.

Comment 5. Many in-line citations are not properly formatted (e.g., citations in lines 129 and 134). These citations should be within parentheses.

**Questions:**

See the weaknesses above.

**Details Of Ethics Concerns:**

Some of the behavioral families crafted by the authors include sensitive descriptors such as “god,” “pregnant woman,” and “serial killer.” I also mentioned this in my review, as these expressions, while I understand the authors’ experimental intent, pose potential ethical and safety concerns. They could inadvertently encourage manipulative or unsafe debating styles. I recommend that the authors replace these terms with neutral, rhetorically defined personas or, at minimum, demonstrate that such controversial families do not persist in the evolutionary process.

---

### Official Review · Reviewer_Y5Zi · 2025-11-03

**Soundness:** 3
**Presentation:** 3
**Contribution:** 3
**Rating:** 6
**Confidence:** 4

**Summary:**

The authors propose a prompt optimization evolutionary algorithm that evolves diverse debate strategies through tournament-style competition, swapping the fitness function to persuasion reward to enforce diversity and subsequently improve LLM generalisation. The authors use Elo ratings score to identify winning prompts.

**Strengths:**

- Clear novel methodology and setup
- The authors show that the argumentative setup pressures transferable skills
- The authors evaluated their methodology across multiple model sizes, up to 72B

**Weaknesses:**

- It is not entirely clear to me if persuasion training is significantly outperforming the truthful strategy based on accuracy results presented in the paper - generalization gaps performance metrics lack interpretability and absolute accuracy appear less significant for the 72B model
- How much does the persuasion fitness function influence the factual accuracy of answers? Can persuasion-based optimization be misaligned with truthful outputs?

**Questions:**

- What does 400 correspond to in Eq 2-3?

**Details Of Ethics Concerns:**

/

---

### Official Review · Reviewer_cCWh · 2025-11-03

**Soundness:** 2
**Presentation:** 3
**Contribution:** 1
**Rating:** 2
**Confidence:** 4

**Summary:**

The paper proposes a framework for evolving  debate strategies in LLMs by optimizing either for persuasion (winning debates) or truth (correctness on the underlying task). The authors show that persuasion-optimized prompts achieve smaller train–test generalization gaps and greater diversity than truth-optimized ones. They claim this suggests competitive pressure fosters more generalizable reasoning.

**Strengths:**

- **[Clear experimental framing]** The comparison between persuasion and truth objectives under fixed debate protocols is well-defined.

- **[Relevance]**: Addresses generalization and overfitting in LLMs is an important problem.

- **[Experiments with large and small models]** The experiments cover multiple model sizes (7B–72B), although these experiments only cover a single dataset (QuALITY) and a single model family (Qwen-2.5).

- **[Clarity/Transparency]**: The authors' methodology and settings are clearly outline and the Appendix details (prompts, tournament design, etc.) are well documented.

**Weaknesses:**

- **[Limited novelty of optimization scheme]**  The authors have two main contributions; an optimization framework and observations made with the context of the optimization framework. The optimization framework itself lack novelty as the proposed “evolutionary" approach is largely an aggregation of existing methods (prompt evolution + Quality/Diversity + Elo tournaments). No new optimization or algorithmic principle is introduced.

- **[Observational/causal ambiguity]**  Regarding the authors second main contribution (observations about the effects of diversity), it remains unclear whether improved generalization stems from the persuasion objective itself or artifacts such as response length, verbosity, or mutation randomness. Given that there many moving pieces affecting the generalizability of the models and the fact that the authors primary contribution is an empirical observation about X thing causing Y effect, I would have expected to see more thorough experiments isolating the underlying relationship.

- **[Weak ablations]** The above issues is the results of incomplete experimentation/analysis within the papers. For example, no experiments isolate the effect of QD vs random mutation, or control for stylistic differences between persuasion and truth prompts.

 - **[Surface-level diversity claim]**  The reported “diversity” is measured via embedding distance, not semantic or strategic difference, and may arise trivially from prompt mutation or simply from the effects of longer responses on embedding distance.

 - **[Lack of statistical significance]** Some of the authors claims do not appear to stem from results which are statistically significant. For example, no confidence interval is given for diversity (Fig 3), for accuracy we really only see a difference in model performance on the train set (on the test set it appears that Persuasion and Truth result in nearly identical accuracy, expect for 32B model, which some how is doing worse than the 7B model . . . ).

- **[Limited experimental scope]**  While the authors do study different model sizes and datasets, results are limited to reading comprehension (QuALITY) using one model family (Qwen 2.5). Broader reasoning or truth-seeking tasks would strengthen the claim.

- **[Limited optimization framework]** The authors attempt to make a broad sweeping claim about the relationship between optimizing for persuasiveness vs correctness, yet they only use a simple prompt-based optimization scheme. To make such a broad claim I would have expected to see multiple paradigms used.

- **[Results not justifying claims]** When viewed in aggregate, the limited nature of the authors experiments and analysis do not justify the broad claims that they make. While the abstract makes the paper and its finding sound quite grand, it is in reality, a fairly shallow and incomplete investigation. I would have found it helpful to see more both more thorough and more broad experiments that get at the heart of the authors' claims about the relationship between generalization and persuasiveness.

**Questions:**

See Weaknesses

---

### Official Review · Reviewer_Apub · 2025-11-05

**Soundness:** 2
**Presentation:** 3
**Contribution:** 2
**Rating:** 2
**Confidence:** 4

**Summary:**

This paper introduces a Quality-Diversity-based prompt optimization framework (DebateQD) which iterates debate strategies for LLMs through prompt evolution. The authors propose two optimization swappable objectives: persuasion, to convince a judge regardless of truth, and  truth, to conclude collaborative correctness. Through empirical experiments across three model scales of Qwen, they show that persuasion-optimized strategies achieve smaller train-test generalization gaps in reading comprehension tasks for most test conditions. For the main contribution, they claim that the proposed framework produces optimized prompts that elicit better generalization, compared to truth-based prompt optimization.

**Strengths:**

The demonstration of applying prompt optimization strategies instead of SFT provides a useful data point about debate training.

**Weaknesses:**

Although the paper shows empirical improvements on train-test generalization, the contribution is limited due to some obvious unexplained design choices and missing work.
- QD category and seed prompt design - The paper mainly uses 7 distinctive categories along with some seed prompts that fit the category in the debate protocol. However, the paper does not justify why these categories are picked and how prompts are chosen? How would choice of these categories and prompt impact the effectiveness of technique? Would missing categories or specific seed prompt reduce the train-test generalization as well?
- Insufficient analysis on robustness to judge and mutator - This paper uses the Qwen the same model as mutator and judge to avoid confounder. However, the success of this technique is strongly dependent on (1) how prompts are mutated through evaluation (2) how judges interpret and decide the persuasion statements. Analysing this frameworks’ robustness to model family and reasoning capability is critical for the contribution to stand. Some additional study of judges and mutators’ requirements is also going to help theorize the technique. Some other questions also remain unstudied, such as the impact of judge biases.
- Only one domain coverage: Evaluating this technique in only one domain is insufficient evidence for generalizing reasoning. Similar experiments should be considered to be done on other domains such as math problems, coding tasks etc.
- Insufficient compute and cost limitation discussion: The technique proposed requires more inference through iterative cost. The incurred complexity in computation cost and time should be discussed in the limitation section.

**Questions:**

Need more interpretation of results: Why is Truth optimization more advantageous in certain cases? In different domains and test cases, how would persuasion optimization vs truth optimization be selected?

---

### Meta-Review · Area_Chair_oqAs · 2026-01-07

**Summary:**

Reviewers were concerned about insufficient experiment and analysis of the proposed method of the paper. In addition,  reviewers were concerned about the lack of ablations and statistical significance. In addition, reviewers were concerned about the arbitrary design choices of the approach, like the choice of the 7 categories of prompt evolution. Finally reviewers were concerned about the limited experimental evaluation and novelty of the approach.

**Reviewer Concerns:**

The authors did not provide a rebuttal, so reviewer concerns are not addressed.

**Reviewer Scores:**

The reviewer scores are not changed as no rebuttal was given.

---

### Decision · Program_Chairs · 2026-01-26

Reject